# Effects of an Agro-Healing Horticultural Intervention on Stress, Self-Esteem, and PANSS Scores in Inpatients with Schizophrenia: A Quasi-Experimental Study

**DOI:** 10.3390/healthcare14010132

**Published:** 2026-01-05

**Authors:** Sang-Mi Lee, Eun-Ju Song, Sun-Jin Jeong, Jiwon Moon

**Affiliations:** 1National Institute of Horticultural and Herbal Science, Rural Development Administration, Jeonju 54875, Republic of Korea; sangmilee@korea.kr (S.-M.L.); sunjin75@korea.kr (S.-J.J.); moonjw85@korea.kr (J.M.); 2Department of Nursing, Wonkwang University, Iksan 54538, Republic of Korea

**Keywords:** horticultural therapy, schizophrenia, self-esteem, stress

## Abstract

**Background:** Developing and implementing diverse interventions is imperative for addressing schizophrenia in the context of deinstitutionalization policies. This study evaluates the effectiveness of an agro-healing horticultural therapy as a psychosocial rehabilitation program for inpatients with schizophrenia. **Methods:** This quasi-experimental study utilizes a non-equivalent control group pre–post design to assess the efficacy of a therapeutic horticulture program for patients diagnosed with schizophrenia. **Results:** This study’s findings supported the initial hypothesis, as the experimental group exhibited a statistically significant reduction in perceived stress, with post-test scores differing significantly between groups (*Z* = −2.11, *p* = 0.035). Hypothesis 2, which examined self-esteem, was rejected because no statistically significant differences were found between groups (*Z* = −0.57, *p* = 0.566). Hypothesis 3 was supported, as the experimental group’s Positive and Negative Symptom Scale (PANSS) scores decreased following treatment, with significant post-treatment differences between groups (*Z* = −3.43, *p* < 0. 001). **Conclusions:** The agro-healing horticultural therapy program in this study effectively reduced stress and PANSS scores among inpatients with schizophrenia. Combining this intervention with medication may enhance rehabilitation outcomes and quality of life for patients afflicted with schizophrenia.

## 1. Introduction

### 1.1. Background

Schizophrenia is one of the most prevalent mental illnesses, characterized by recurrent relapses and a diminished quality of life [1,2]. This diminished quality of life can be attributed to the chronic nature of the illness, which results in impaired cognition, perception, emotion, and behavior, placing significant socioeconomic burdens on families and communities [3]. The growing prevalence of mental illnesses, including schizophrenia, profoundly affects health, social conditions, human rights, and economic outcomes worldwide [4]. Recent psychiatric interventions have placed a major focus on improving the quality of life, aiming to enhance the patient’s ability to lead a satisfactory life despite their illness [5,6]. The success of rehabilitation ultimately contributes to resolving social and economic issues associated with schizophrenia patients; therefore, the development of diverse and effective interventions is required.

Historically, intervention studies for schizophrenia have primarily focused on improving positive symptoms, negative symptoms, and cognitive impairment. However, complementary approaches are increasingly needed to address patients’ overall functioning and quality of life. Among various methods, horticultural therapy, initially adopted by 19th-century American psychiatrist Benjamin Rush, has recently gained significant attention in Asian countries like South Korea, China, and Japan [7,8]. This therapy is characterized by its flexible application in diverse environments, ranging from dedicated indoor spaces to various outdoor settings such as therapeutic gardens or courtyards [3,9]. This adaptability allows participants to engage with plants and experience the natural changes of the four seasons. Research indicates that horticultural therapy-based interventions are more effective than general interventions for schizophrenia, specifically in reducing stress, psychiatric symptoms, and enhancing self-esteem [7,10]. Building upon this evidence, this study aims to implement and evaluate an agro-healing horticultural therapy program for inpatients with schizophrenia.

Horticultural therapy is conventionally defined as a therapeutic process that involves diverse activities primarily utilizing plants and gardening—such as direct observation, olfactory perception, and manual cultivation during the process of harvesting crops [8]. This approach has been noted for its cost-effectiveness and potential to generate favorable outcomes in domestic settings [7,11]. In recent years, this concept has been broadened into agro-healing horticultural therapy, a more comprehensive and multidisciplinary approach [12,13]. Agro-healing utilizes the entire farm and rural environment to promote psychological, social, cognitive, and physical health, encompassing elements beyond plants, such as livestock raising, forestry, and rural cultural resources [3]. These activities leverage the unique atmosphere of farms and rural landscapes to facilitate both direct and indirect cultural and emotional exchanges [12,13].

Agro-healing horticultural therapy has been specifically developed to assist diverse populations, including individuals requiring support due to physical or emotional illness, socially vulnerable groups, marginalized communities, those with mental disorders requiring medical or social treatment, and the elderly, thereby aiding in the restoration of physical, mental, and social well-being [14]. The specific intervention implemented in this study was an agro-healing approach focused on horticultural activities. This structured therapy was designed to enhance physical, cognitive, and mental health by integrating specific gardening tasks within a therapeutic framework based on interaction with the natural environment.

The curriculum integrated into an educational component, allowing participants to learn about the characteristics and benefits of crops consumed by humans through the processes of sowing, growth, and harvesting, along with a healing aspect facilitated by communion with nature [15]. In other words, it was designed as more than physical activity, incorporating farm resources, including crops and the landscape, to reduce anger and fatigue while fostering self-efficacy through agricultural activities [13].

Horticultural plants have been shown to stabilize the autonomic nervous system and reduce stress [16]. Furthermore, agro-healing horticultural therapy has been demonstrated to be effective in reducing stress across all age groups [17]. Given their applicability to individuals experiencing high levels of stress and those with treatable mental disorders [18], these activities are considered suitable for patients with schizophrenia, potentially leading to improvements in their overall psychosocial functioning. Stress is conceptually defined as a dynamic process involving the psychological and physiological reactions of an organism to perceived threats or challenges, leading to a state of internal disequilibrium [19]. These responses are readily quantifiable by validated psychological and physiological measures. Self-esteem refers to an individual’s overall subjective evaluation of their own worth [20], which is also readily measurable using standardized self-report tools. These variables are highly relevant to schizophrenia, as chronic stress can trigger relapses, while low self-esteem impairs social functioning.

Individuals with schizophrenia often experience low self-esteem due to challenges in social adaptation and the impact of stigma [21]. Low self-esteem can be addressed through intervention, which is also a goal of agro-healing horticultural therapy [22]. Horticultural therapy has been shown to significantly enhance confidence and self-esteem, foster hope for the future, and improve interpersonal relationships and social skills among patients with schizophrenia [3].

Agro-healing horticultural therapy has also been employed as an auxiliary rehabilitation therapy, demonstrating efficacy in reducing psychopathological symptoms, anxiety, and depression, while improving social functioning [12,18]. Additionally, this approach has proven effective in vocational adaptation training for patients with mental illnesses facing economic hardship, as it enhances psychological stability, self-confidence, and a sense of accomplishment [16]. Furthermore, the simple and accessible nature of these activities stimulates interest in individuals with mental illness [7], and is considered beneficial for patients with schizophrenia exhibiting both positive and negative symptoms.

The Positive And Negative Symptom Scale (PANSS) is widely used in clinical research and practice to monitor symptom changes in patients with schizophrenia and to evaluate treatment response, including comparing symptom differences between patient groups [23]. The PANSS is the most commonly used measure in horticultural therapy research and has demonstrated proven efficacy in previous studies [2,3,7]. Participants engage in physical activities by handling soil and observing crop growth, which is expected to positively affect patients with schizophrenia, particularly those with predominant negative symptoms.

### 1.2. Purpose and Hypothesis

The purpose of this study is to investigate the effects of an agro-healing horticultural therapy program on stress, self-esteem, and PANSS scores in inpatients with schizophrenia.

The hypotheses for the present study are as follows:

**Hypothesis** **1.**
*The experimental group participating in the agro-healing horticultural therapy program will show greater improvement in perceived stress compared to the non-participating control group.*


**Hypothesis** **2.**
*The experimental group participating in the agro-healing horticultural therapy program will show increased self-esteem compared to the non-participating control group.*


**Hypothesis** **3.**
*The experimental group participating in the agro-healing horticultural therapy program will show greater improvement in PANSS scores compared to the non-participating control group.*


## 2. Methods

### 2.1. Participants

The participants in this study were patients diagnosed with schizophrenia according to the Diagnostic and Statistical Manual of Mental Disorders, Fifth Edition and admitted to a psychiatric hospital located in J province, South Korea.

Two wards with similar functional levels among the hospitalized patients were selected. To prevent treatment contamination and avoid interference, Ward A was designated as the experimental group and Ward B as the control group, following discussions with the psychiatrists and psychiatric nurses. The initial participant pool was recruited from among all patients admitted to the two designated wards during the recruitment period (e.g., from start date to end date), after the attending psychiatrists and head nurses of the respective wards identified patients meeting the selection and exclusion criteria. The two wards were located in separate buildings, physically distant from each other. Both wards conducted similar programs—such as social skills training, exercise program, music therapy, and art therapy—with the control group limited to these activities. Differences in walking schedules further minimized opportunities for participants to meet.

The sample size for this study was calculated using G-Power 3.1.9 (Heinrich Heine University, Dusseldorf, Germany). The F-test for Analysis of Covariance (ANCOVA) was used for the estimation. A power analysis using an effect size (f) of 0.51 (Cohen’s criterion for a large effect) [22] yielded a minimum requirement of 21 participants per group to achieve a significance level (α) of 0.05 and a statistical power (1-β) of 0.80. Considering potential dropouts, 31 participants each were recruited for the experimental and control groups. Due to discharge or treatment at other hospitals, four participants from the experimental group and seven from the control group did not complete the post-test. The final sample size comprised 26 participants in the experimental group and 23 in the control group. The average age of participants in this study was 52.9 years, and the male-to-female participation ratio was 51:49.

Applicants who expressed interest were evaluated according to predefined inclusion and exclusion criteria. The specific selection criteria for the participants were as follows:Patients hospitalized for at least three months with stable symptoms;Individuals without a dual diagnosis of other mental disorders, such as substance-related disorders or neurocognitive disorders;Individuals able to read and respond to questionnaires, with no communication impairments;Individuals who received an explanation of the research, understood its purpose, and voluntarily agreed to participate.

The specific exclusion criteria for the participants were as follows:Individuals who have difficulty in program execution and communication challenges due to cognitive impairment.Individuals exhibiting psychotic symptoms severe enough to impede cooperation with program execution.Individuals with a risk of violent behavior, self-harm, or harm to others resulting from their symptoms.

### 2.2. Procedure

#### 2.2.1. Design

This quasi-experimental study employed a non-equivalent control group pre–post design to assess the efficacy of an agro-healing horticultural therapy program for patients with schizophrenia.

#### 2.2.2. Data Collection

Data collection for this study took place from 23 August to 25 October 2024, and the program was conducted on a weekly basis. The authors visited the psychiatric hospital and obtained permission from the hospital director and the attending psychiatrist. The study was conducted by four agricultural researchers and one nursing professor. Before starting, the researchers met with the attending psychiatrist to discuss participant selection and program content. Confidentiality was assured to both the experimental and control groups before the program began, and pre-tests were administered on the start date. Post-tests were conducted at the program’s conclusion. Perceived stress and self-esteem were assessed through self-report questionnaires, while the PANSS was administered with the assistance of two psychiatrists and two psychiatric head nurses. The pre-test was administered immediately prior to the initiation of the intervention program, and the post-test was conducted immediately following the conclusion of the final session. All questionnaire assessments for both the experimental and control groups were conducted concurrently, in the respective ward program rooms, following identical procedures.

#### 2.2.3. Ethical Considerations

This study was conducted following the principles of the Helsinki Declaration and approved by the Institutional Review Board of Jeonbuk National University (Approval No.: JBNU 2024-05-038-001). Prior to participation, the attending two psychiatrists, who are specialists in psychiatry, clinically assessed the participants’ decision-making capacity. The attending psychiatrists were responsible for making the ultimate determination regarding the individual’s eligibility and capacity to consent, taking into account both clinical stability and ethical competence. This assessment was conducted to ensure that the participants had a comprehensive understanding of the study’s purpose, procedures, risks, and benefits, and were capable of providing voluntary consent. The evaluation determined that all participants were capable of providing voluntary informed consent. The procedure for confirming decision-making capacity was conducted in accordance with ethical guidelines, including the Declaration of Helsinki, to maximize respect for the autonomy of psychiatric patients, who are considered a vulnerable population.

Participants who expressed willingness to join the study were provided with detailed explanations of all relevant information, including the study’s purpose, procedures, and schedule. Written consent was obtained from those who agreed to participate. Participants were assured that there would be no adverse effects or risks associated with participating in this study and that the collected data would be used solely for research purposes. They were also informed that they could withdraw from the study at any time without facing any negative consequences.

#### 2.2.4. Agro-Healing Horticulture Therapy

The agro-healing horticultural therapy program for patients with schizophrenia in this study was grounded in the care farm concept proposed by Hine et al. [24]. This concept involves providing care services—encompassing health recovery, social rehabilitation, and educational activities—to individuals in need of healing, with various manifestations depending on the participant group and specific situation [24]. Given that horticultural therapy programs require the design and implementation by qualified professionals [25], the intervention was designed through a multidisciplinary collaboration. Specifically, four agricultural researchers affiliated with the Korean Rural Development Administration developed horticultural sessions. To ensure the sessions were appropriately tailored to focus on the specific symptoms of schizophrenia patients, two psychiatrists and a professor of psychiatric nursing also participated in the comprehensive design process. This specialized, multidisciplinary framework forms the foundation of the program, which focuses on the therapeutic benefits of the plant life cycle—from sowing to harvest. The program delivery was subsequently entrusted to four agricultural researchers who served as the facilitators. These individuals possess substantial expertise in horticultural therapy, having each accumulated over ten years of experience in the field. In addition, they have each accumulated a minimum of three years’ experience in planning and implementing these agro-healing programs specifically for patients in psychiatric wards.

Based on the specialized framework, the program was organized into 10 weekly sessions (see Table 1), taking place on Fridays over a 10-week period. Each session, conducted at the psychiatric hospital, lasted 60–90 min and followed a structure including an introduction, development, and conclusion. All sessions were conducted at a designated therapeutic garden facility located within the psychiatric hospital. The facility included an outdoor garden and vegetable patch, complete with seating and rest areas for participants. To ensure continuity during adverse weather conditions, an indoor space was also available. The program was implemented during the fall season. All sessions were conducted outdoors under consistently favorable weather conditions, with ambient temperatures maintained at approximately 25 degrees Celsius.

The agro-healing program was designed as a time-based intervention to allow participants to experience the entire life cycle of horticultural plants, including propagating, caring, harvesting, and utilizing. Considering the participants’ ability to sustain engagement and the institution’s capacity to manage a large number of participants, the program was structured over 10 sessions. Each session included approximately 40 min of gardening activities as the primary component, 15–20 min of active sun exposure, and 20–30 min for completing activity sheets and sharing reflections.

The control group did not participate in the main agro-healing intervention. Instead, they participated in a standard, ward-based psychiatric rehabilitation program that included music, exercise, and art therapies, as well as social skills training. The program was structured to meet four days a week to ensure equivalence in contact time. Both groups participated in the program for 50 min daily, Monday through Thursday. The control group’s weekly schedule was fixed: music therapy on Monday, exercise therapy on Tuesday, art therapy on Wednesday, and social skills training on Thursday. The Friday sessions varied between the two groups. Only the experimental group received the complementary agricultural healing horticultural therapy session on Friday. Thus, over the 10-week period, the control group received a total of 40 sessions (four times a week), while the experimental group received 50 sessions (five times a week). Importantly, the standard ward program for the control group did not include exposure to agricultural activities, which were a core element of the experimental program.

Each session was organized as follows:

The first session, titled “What Kind of Seed Are You?”, was designed to help participants identify their strengths, which were then compared to the potential and advantages of the seeds to be sown. Participants were guided through an immersive experience, interacting with the soil and its texture during seed-sowing. Repetitive sowing activities also helped reduce excessive vigilance toward external stimuli and promoted immersion by encouraging focus on the present moment.

Session 2, titled “Gardening Together,” aimed to highlight the interdependence of plants and human collaboration through the shared gardening experience. Participants first worked collaboratively to plant seedlings in the designated garden while learning about the symbiotic relationships among plants. They immersed themselves in the experience, focusing intently on the sensation in their hands and each movement as they removed seedlings from trays and transplanted them into the garden plot. After completing the work, team members exchanged words of appreciation and gratitude, fostering interpersonal relationships and emotional connections. Finally, through reflection on the planting activity and expression of positive sentiments, participants were guided to experience positive emotions.

Session 3, titled “Consideration for Being Together,” focused on relationship building. Participants were encouraged to reflect on the concept of “spacing” in human relationships, emphasizing the importance of respect and care for one another. Extracting seedlings from trays and planting them at the appropriate spacing necessitated repetitive, precise hand movements, fostering concentration. After completing the activity, participants engaged in a reflective exercise, contemplating the elements of respect and compromise necessary for fostering healthy relationships, guided by the question, “What distance should be maintained between myself and others to sustain healthy relationships?”

Session 4, titled “I Will Protect You,” was designed as a socialization intervention aimed at cultivating relationship awareness. The session focused on protecting plants using eco-friendly pest control activities. Participants engaged in hand movements and tool use by dividing and diluting a concentrated pest control solution made from natural ingredients and then spraying it directly onto the plants. This activity allowed them to experience the sense of forming a relationship with the plants through protection and to recognize an emotional connection in caring for and protecting another being. Subsequently, participants were guided to reflect on their roles in self-other relationships using questions such as, “In my relationships with others, how do I give and receive help?”

Session 5, titled “Lean on Me When You Are Tired and Weary,” was designed to emphasize relationship recognition. The activity involved setting up supports and string guides to facilitate the growth of tomato plants, which are fruit vegetables, and cucumbers, which are vine plants. Learning and applying techniques such as pruning, securing tomato stems to support, and tying cucumber vines to strings required careful attention to subtle hand movements, fostering attention. Following the activity, participants reflected on their own support systems and shared insights about the roles they play in their relationships with others. This introspection was guided by questions such as, “Have I ever been a recipient of such support from another person?” or “Have I ever been a source of support for another individual?”

The sixth session was held under the theme “Nourishment for My Heart”. Participants measured and applied the appropriate amount of fertilizer to the garden bed according to the type and condition of the plants, fostering concentration in the activity. They reflected on the mental nourishment required for healthy growth by recognizing their potential and strengths and were encouraged to express hope for the future of both the plants and them.

In the seventh session, titled “I Grew It Myself,” participants were guided through the meticulous process of selecting, trimming, and arranging flowers. Building on the stem-cutting and flower-harvesting activities from Session 2, participants created floral arrangements by inserting the flowers into floral foam. They were encouraged to cultivate a sense of accomplishment and express pride through their arrangements and cards. After completing their floral arrangements, participants presented their creations to themselves, wrote messages of encouragement and support on cards, envisioned a positive future, and took time to recognize and express their strengths.

In the eighth session, which was entitled “Soaked in Fragrance,” participants engaged with the sensory experience through tactile, olfactory, and thermal stimuli. They engaged in activities such as harvesting herbs, trimming tea leaves, and brewing tea within the designated time frame, fostering nature immersion in the present moment. Harvesting and preparing their own tea instilled a sense of achievement among participants. Inhaling the aroma of herbal tea and savoring the brew supported emotional stability and positive moods, while also helping participants recognize and express their own distinctive scent in a positive manner.

Session 9, titled “We Will Bloom Like Flowers,” focused on achievement and strengths. Participants created cards using pressed flowers saved from previous flower decoration activities, immersing themselves in the sensory experience by meticulously arranging and carefully attaching the flowers to the cards. They were encouraged to cultivate a sense of accomplishment and express pride through their creations. Participants also wrote messages of encouragement and support on the cards, envisioned a positive future, and took time to recognize and express their strengths.

The final session concluded with an event themed “Enjoying a Garden Party Together”. Participants concentrated on the preparation process and the party itself, decorating tables and preparing refreshments using vegetables and flowers harvested from the garden. Harvesting the plants they had carefully tended and using them to prepare their own party imparted a profound sense of achievement. Participants praised one another, reflected on their growth during the program, and expressed pride. Expressing satisfaction and gratitude for completing the session encouraged a positive outlook on life, which further extended to reflections on hope for the future and the meaning of life.

The specific activities and key factors of the agro-healing horticulture are outlined in Appendix A.

### 2.3. Measurements

#### 2.3.1. Perceived Stress Scale (PSS)

Perceived stress refers to the degree of stress that individuals actually feel and interpret. This study utilized the Perceived Stress Scale (PSS) developed by Cohen et al. [26] and adapted into Korean by Lee et al. [27]. The PSS measured participants’ perceived stress over the past month using 10 items. It has a unidimensional (one-factor) structure and is scored on a 5-point Likert scale ranging from 0 (“never”) to 4 (“very often”). Higher scores indicated higher levels of perceived stress. In Harris et al.’s [28] study, Cronbach’s s α was 0.89; in the present study, Cronbach’s α was 0.75.

#### 2.3.2. Self-Esteem

This study used the Self-Esteem Scale developed by Rosenberg [20] and translated into Korean by Lee and Won [29] to measure self-esteem. This scale comprises 10 items: five assessing positive self-esteem and five assessing negative self-esteem. It is generally regarded as having a unidimensional structure. Items are scored on a 5-point Likert scale ranging from 1 (“strongly disagree”) to 5 (“strongly agree”), with higher total scores indicating a higher self-esteem. In Lee and Won’s [29] study, Cronbach’s α was 0.89; in Han et al. [30], Cronbach’s α was 0.90; in the present study, Cronbach’s α was 0.75.

#### 2.3.3. Positive and Negative Symptom Scale (PANSS)

This study used the PANSS developed by Kay, Fiszbein, and Opler [31] and standardized in Korean by Yi et al. [32]. It consists of 30 items: seven assessing positive symptoms, seven assessing negative symptoms, and 16 assessing general psychopathology. It is based on a three-factor structure. Symptoms are scored on a 7-point Likert scale ranging from 1 (“none”) to 7 (“most severe”), with higher scores indicating more severe psychotic symptoms. Example items include Hallucination (Positive), Emotional Withdrawal (Negative), and Anxiety (General Psychopathology). The PANSS was designed to evaluate positive, negative, and general psychopathological symptoms in patients with schizophrenia. Positive symptoms include hallucinations, delusions, confusion, and aggression, while negative symptoms include social withdrawal, emotional blunting, loss of motivation, and impoverished speech. General psychopathological symptoms include anxiety, tension, guilt, and a sense of injustice. Regarding reliability, Yi et al. [32] reported Cronbach’s α of 0.73 for positive symptoms, 0.84 for negative symptoms, and 0.74 for general psychopathology. The PANSS demonstrated a total Cronbach’s α of 0.876 in the study by Kim et al. [33]. In this study, Cronbach’s α was 0.71 for positive symptoms, 0.81 for negative symptoms, and 0.89 for general psychopathology.

### 2.4. Data Analysis

Statistical analysis was conducted using SPSS version 26.0 (SPSS Inc., Chicago, IL, USA). Descriptive statistics were used to determine the general characteristics of the participants, providing the mean and standard deviation. The homogeneity of general characteristics between the experimental and control groups was assessed using the chi-squared test. The Mann–Whitney U test was applied to evaluate the pre-experimental homogeneity of the dependent variables across the two groups. This was necessary because the Kolmogorov–Smirnov normality showed significance levels below 0.05, indicating that the assumption of a normal distribution was not met. Pre- and post-test scores of the experimental and control groups were compared using the ANCOVA.

## 3. Results

### 3.1. Homogeneity Test of General Characteristics

Table 2 presents the general characteristics of the participants, with no statistically significant differences between the experimental and control groups.

Homogeneity testing of the dependent variables between the experimental and control groups revealed significant disparities in perceived stress (Z = −2.08, *p* = 0.037), self-esteem (Z = 2.30, *p* = 0.021), and PANSS (Z = −4.25, *p *< 0.001), indicating that the groups were not homogeneous. However, the Negative Symptom subscale of the PANSS showed no statistically significant difference (Z = −1.44, *p* = 0.151). The results of the homogeneity test for the dependent variables are presented in Table 3.

### 3.2. Effects of the Agro-Healing Intervention

For Hypothesis 1, which predicted that the perceived stress scores would significantly decrease in the experimental group compared to the control group, was supported. The Analysis of Covariance confirmed that the adjusted post-test score for the experimental group was significantly lower than that of the control group, supporting the intervention effect (F = 4.23, *p* = 0.021).

Hypothesis 2, which predicted that the self-esteem scores would significantly increase in the experimental group compared to the control group, was not supported. Although the experimental group’s self-esteem increased slightly (from 31.87 to 32.00) while the control group’s decreased (from 35.96 to 34.11), no statistically significant difference was found, leading to the rejection of the hypothesis (F = 0.22, *p* = 0.638).

Hypothesis 3, which predicted that the PANSS scores would significantly decrease in the experimental group compared to the control group, was supported. The Analysis of Covariance confirmed that the adjusted post-test score for the experimental group was significantly lower than that of the control group, supporting the intervention effect (F = 9.60, *p* = 0.003).

Table 4 presents the pre- and post-test scores of the two groups and the results of the hypothesis testing.

## 4. Discussion

This study confirmed the hypothesis that the agro-healing horticultural therapy program produced favorable changes in stress and PANSS scores among patients diagnosed with schizophrenia.

Hypothesis 1, which predicted a decrease in the experimental group’s perceived stress scores, was supported. Consistent with the findings of similar studies, including Kim, Choi, and Sung [16], which measured perceived stress, and Eum and Kim [34], which assessed stress responses, patients with schizophrenia who participated in horticultural therapy exhibited a statistically significant decrease in their stress scores. Limited prior research has examined stress as a measurement variable in horticultural therapy for patients with schizophrenia; most studies have instead primarily focused on anxiety and depression [3,7]. Therefore, this study suggests that stress can be considered a psychological variable in future research. The agro-healing horticultural therapy program was delivered outdoors on sunny days. Each session featured diverse physical activities and interpersonal interactions, contributing to stress reduction. Patients with schizophrenia are more vulnerable to stress than the general population or those with other mental disorders. Due to the nature of the illness, they also experience chronic stress and reduced capacity for effective coping [35]. Furthermore, hospitalized patients with schizophrenia face behavioral limitations caused by their restricted environment and exposure to various stressors. Consequently, a range of interventions is required to help them identify stressors and develop effective coping mechanisms, which may positively impact their lives after discharge.

Hypothesis 2, which predicted a significant increase in self-esteem scores in the experimental group, was rejected. These findings contradict those of previous horticultural therapy studies for individuals with mental illness, such as the research by Subagyo and Wahyuningsih [36] and Song et al. [37], which reported significant improvements in self-esteem using similar instruments and interventions. Previous research suggests that achievement-oriented tasks within horticultural therapy, such as pest control, fertilizing, and watering, enhance self-esteem by fostering goal attainment and improving concentration [36]. The lack of a significant effect in the current study may be attributed to several factors. First, the agro-healing program, while developed with specialists, may not have provided sufficient opportunities for mastery experiences tailored to the specific cognitive and emotional profiles of the participating patients with schizophrenia. Second, the duration of the intervention (implied but not stated in this section) may have been insufficient to produce a measurable change in a complex, relatively stable psychological construct like self-esteem. Considering the critical role of self-esteem in the recovery and rehabilitation outcomes for individuals with schizophrenia, where low self-esteem is linked to an increased risk of relapse [38] and hinders environmental adaptability [1], future research is necessary. Subsequent studies should focus on developing horticultural therapy programs that carefully consider the severity of schizophrenia symptoms and are specifically designed to maximize experiences of achievement and competence [39], thereby targeting this key psychological variable effectively.

Hypothesis 3, which predicted a significant decrease in PANSS scores in the experimental group compared to the control group, was supported. The experimental group demonstrated a marked reduction in psychopathological symptoms, while the control group showed almost no change. This result supports the established efficacy of agricultural therapeutic horticultural therapy in treating schizophrenia symptoms, as demonstrated in previous studies utilizing the PANSS [3,7,34]. Even in studies that assessed psychosis using the Brief Psychiatric Rating Scale instead of the PANSS, a significant improvement in psychotic symptoms was found in the experimental group of patients with schizophrenia who received horticultural therapy [10,40]. A comprehensive evaluation of all subdomains (positive, negative, and general psychopathological symptoms) in the present study revealed substantial improvement in the experimental group. This suggests that the agro-healing program effectively addressed the multidimensional nature of schizophrenia symptoms. Specifically, the comprehensive improvement observed across the PANSS subscales provides a deeper understanding of the intervention’s effects.

The significant reduction in positive symptoms is noteworthy and supports research suggesting that the physical and interactive activities inherent in agro-horticultural therapy can facilitate distraction from delusional thinking and enhance reality testing [7,34]. As a non-pharmacological intervention, these effects are interpreted as indirect, contrasting with the direct biological action of medication. Active engagement in tasks such as planting or weeding is demonstrated to serve as a valuable form of behavioral distraction from positive symptoms. This focus on physical activity likely diverts cognitive resources away from internal preoccupations such as delusions or hallucinations. This sustained external focus, facilitated by the stimulating natural environment, may have reduced the intensity and frequency of positive symptoms, as reflected in the reduction in PANSS scores.

Furthermore, the improvement in negative symptoms, which often do not respond to standard pharmacological treatments, highlights the unique potential of this intervention to increase engagement, motivation, and social interaction. Overall, the results highlight the significant potential of agro-horticultural therapy as a non-pharmacological adjunctive intervention to improve psychosocial functioning and quality of life in patients receiving standard pharmacological treatment, emphasizing that it is not a replacement for conventional psychiatric treatment. Psychiatrists and researchers should be interested in developing these novel adjunctive interventions to improve outcomes beyond those achieved with antipsychotic medications alone [6]. Specifically, these results strengthen the hypothesis that the substantial physical activity and interactive activities inherent in the intervention, conducted in an outdoor environment, were effective in promoting positive symptom relief.

The significant improvement in negative symptoms observed in the experimental group is an important finding of this study. While third-generation antipsychotic drugs are effective in treating certain aspects of schizophrenia, previous research indicates that optimal management of negative symptoms often requires the combination of pharmacological treatment with non-pharmacological interventions that encourage physical activity and social engagement [2,3]. The results of the present study support this integrated approach. Consistent with findings that show combined medication and horticultural therapy leads to substantial improvement in negative symptoms [2,34], our agro-healing therapy successfully facilitated patient engagement. The outdoor setting and structured physical activities, such as planting, watering, and weeding, likely contributed to alleviating negative symptoms. Furthermore, the mandatory communication and interpersonal interactions may provide opportunities to improve emotional expression and social withdrawal, as evidenced by participants’ brighter facial expressions—key indicators of reduced emotional flatness and lack of affect. This outcome does not suggest that agro-healing horticultural therapy replaces the fundamental role of antipsychotic medication. Rather, it demonstrates that integrating psycho-social interventions like agro-healing can effectively address chronic and pervasive negative symptoms, thereby enhancing the overall therapeutic effect achieved by standard pharmacological care.

Regarding the general psychopathology subscale of the PANSS, which measured a wide range of symptoms. The experimental group demonstrated a significant reduction in symptoms measured by the general psychopathology, which covers a wide range of issues including anxiety, guilt, depression, motor retardation, uncooperativeness, insight, and impaired judgment. This outcome is crucial as these symptoms significantly impair the quality of life and psychosocial functioning of patients with schizophrenia. The observed improvements are likely due to the multifaceted nature of the agro-healing program. Horticultural therapy encourages active, self-directed activities through engagement with plants, which is known to help prevent the regression of mental illness and foster social skills [34]. Furthermore, the experience of tending and growing plants has been shown to alleviate negative emotions and improve mood by building confidence in one’s ability to care for something and supporting a healthy life [10,41]. In this study, the structured social interactions inherent in the activities played a key role. Conversations during agro-healing horticultural work allowed participants to discover and articulate their personal strengths. Additionally, sharing positive emotions during communal cooking sessions, following the gardening tasks, may have functioned as a powerful reinforcing factor, further contributing to the significant improvements observed in general psychopathological symptoms.

The integration of various activity types within the agro-healing program—combining physical tasks with creative, self-directed engagement (e.g., crafts and creation)—is posited as a key contributor to its observed efficacy. This multimodal design aligns with the contemporary shift in schizophrenia treatment goals. Over the past decade, research emphasis has moved beyond the traditional goal of mitigating only positive symptoms to a focus on alleviating negative and cognitive symptoms and enhancing patient functioning and quality of life [1,42]. The positive findings of this study reinforce the notion that using antipsychotic medications with a reduced adverse event profile, combined with established interventions, is essential to enhance rehabilitation outcomes and improve the quality of life for patients with schizophrenia.

This study’s results must be interpreted with caution due to several methodological constraints. First, the small number of patients included, coupled with the fact that participants were only inpatients from a single facility, limits the generalizability of the findings to diverse populations or settings (e.g., outpatient care). Second, there was a degree of baseline heterogeneity between the experimental and control groups at the pre-exposure time point. Third, the possibility that other simultaneous interventions within the ward (e.g., music therapy, exercise therapy, art therapy, and social skill training) influenced the results cannot be completely ruled out. Finally, the difference in the total intervention dosage is a key limitation. Given that the experimental group received an additional treatment session on Fridays (50 total sessions vs. 40 total sessions), the influence of this increased contact time and attention on the results cannot be entirely discounted. Future studies should address these constraints, particularly by considering designs that strictly equalize the total contact time across both groups.

However, the strength of this study lies in the fact that a horticultural therapist directly administered horticultural therapy to patients with schizophrenia, while a psychiatrist and mental health nurse provided support for symptom monitoring and assessment throughout the entire process. Considering the current reality that mental health professionals (e.g., psychiatrists and nurses) in psychiatric wards seek to implement agro-healing horticultural therapy for community reintegration, but face challenges due to the qualifications and prerequisites required to deliver the program [16], this study may be considered meaningful.

Despite these limitations, the present study offers valuable evidence supporting the efficacy and unique potential of agro-healing horticultural therapy as an adjunctive intervention for schizophrenia. Future research should address these methodological constraints to further validate the findings.

## 5. Conclusions

This study provides preliminary evidence suggesting the potential benefits of the agro-healing horticultural therapy program as an adjunctive intervention for patients with schizophrenia who are currently undergoing pharmacological treatment. Rehabilitation for these patients must be approached from a multidisciplinary perspective, rather than being solely a matter for psychiatry. The agro-healing horticultural therapy program integrates such multidisciplinary elements and may potentially serve as a rehabilitation intervention under deinstitutionalization policies.

However, given the exploratory nature of this research and the constraints inherent to the small sample size and single-facility setting, the findings should be interpreted with caution. Thus, it is recommended that subsequent studies employ a more extensive and diverse sample, encompassing a wider range of regions, institutions, and settings, to validate these initial findings.

## Figures and Tables

**Table 1 healthcare-14-00132-t001:** Agro-healing horticultural therapy program.

Session	Classification	Horticultural Activity
1	Breeding	Sowing vegetables
2	Breeding	Transplanting seedlings of flowering plants and herbs
3	Breeding	Transplanting plants shown in Session 1
4	Management	Pest and disease control
5	Management	Pruning
6	Management	Fertilization
7	Utilization	Floral arrangements
8	Utilization	Making herbal tea
9	Utilization	Pressed flower decoration
10	Utilization	Garden party

**Table 2 healthcare-14-00132-t002:** Homogeneity Test of Participants’ Characteristics (N = 49).

Characteristics	Categories	Exp. (n = 26)	Cont. (n = 23)	*x* ^2^	*p*
		n (%)	n (%)		
Age	20–30 s	3 (11.5)	3 (13)	0.43	0.807
	40–50 s	17 (65.4)	13 (58.5)		
	60 s	6 (23.1)	7 (30.5)		
Sex	Male	12 (46.2)	13 (56.5)	0.53	0.469
	Female	14 (53.8)	10 (43.5)		

*Note.* Exp., experimental group; Cont., control group.

**Table 3 healthcare-14-00132-t003:** Homogeneity Test of Dependent Variables Between Groups (N = 49).

Variables	Pretest	Pretest	*Z*	*p*
	Exp. (n = 26)	Cont. (n = 23)		
	M ± SD	M ± SD		
Stress	17.67 ± 5.09	19.08 ± 4.03	−2.08	0.037
Self-esteem	31.87 ± 4.13	35.96 ± 6.39	2.30	0.021
PANSS	92.69 ± 13.32	73.57 ± 12.26	−4.25	<0.001
Positive symptom	22.92 ± 4.91	18.43 ± 4.20	−3.00	0.003
Negative symptom	21.35 ± 6.14	19.00 ± 4.58	−1.44	0.151
Gen.	48.42 ± 6.69	36.13 ± 6.77	−4.68	<0.001

*Note.* Exp., experimental group; Cont., control group; Gen., General psychopathology.

**Table 4 healthcare-14-00132-t004:** Comparison of Dependent Variables Between Groups at Post-test (N = 49).

Variable	Groups	Pretest	Posttest	Difference	F *	*p* *
		M ± SD	M ± SD	M ± SD		
Stress	Exp. (n = 26)	17.67 ± 5.09	15.74 ± 5.50	−1.93 ± 6.16	4.23	0.021
	Cont. (n = 23)	19.08 ± 4.03	17.54 ± 3.77	−1.53 ± 3.89		
Self-esteem	Exp. (n = 26)	31.87 ± 4.13	32.00 ± 4.82	0.13 ± 3.65	0.22	0.638
	Cont. (n = 23)	35.96 ± 6.39	34.11 ± 7.71	−1.85 ± 6.83		
PANSS	Exp. (n = 26)	92.69 ± 13.32	80.46 ± 14.21	−12.23 ± 10.66	9.60	0.003
	Cont. (n = 23)	73.57 ± 12.26	74.52 ± 12.84	0.96 ± 6.42		
Positive symptom	Exp. (n = 26)	22.92 ± 4.91	19.35 ± 5.32	−3.58 ± 3.81	6.71	0.013
	Cont. (n = 23)	18.43 ± 4.20	18.44 ± 4.3.3	0.00 ± 2.70		
Negative symptom	Exp. (n = 26)	21.35 ± 6.14	19.12 ± 5.24	−2.23 ± 2.61	7.95	0.007
	Cont. (n = 23)	19.00 ± 4.58	18.96 ± 4.62	−0.04 ± 1.97		
Gen.	Exp. (n = 26)	48.42 ± 6.69	42.00 ± 7.57	−6.42 ± 5.32	11.43	0.001
	Cont. (n = 23)	36.13 ± 6.77	37.13 ± 7.12	1.00 ± 3.30		

*Note.* Exp., experimental group; Cont., control group; Gen., General psychopathology. * The F and *p* values were derived from the Analysis of Covariance (ANCOVA) controlling for Pretest scores. The *p* value specifically indicates the statistical significance of the difference between the Experimental and Control groups at the Posttest.

## Data Availability

The data presented in this study are available on request from the corresponding author. The data are not publicly available due to ethical restrictions; the datasets generated and analyzed in the study are not publicly available and cannot be shared.

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
