# Peer review of "Effects of an Agro-Healing Horticultural Intervention on Stress, Self-Esteem, and PANSS Scores in Inpatients with Schizophrenia: A Quasi-Experimental Study"

_healthcare, 2026, doi:10.3390/healthcare14010132_

Round 1
Reviewer 1 Report
Comments and Suggestions for Authors
Thank you very much for this interesting article and your reserach about this important topic. To optimize this article, I would recommend following steps:
1.) Include short definitions of horticultural therapy and agro-healing horticultural therapy in the Introduction-chapter.
2.) Further develop the links and relationships between stress, self-esteem and schizophrenia in the Introduction-chapter and offer a short definition of stress and self-esteem (its meaning and the possibilities to measure it).
3.) Also in the Introduction-chapter, you write, that your aim was as follows: "The aim of the agro-healing program was to promote health restoration and rehabilitation through agricultural activities. ". Thus, it would be important to explain the connections between health restoration (and its promotion) and rehabilitations, stress, self-esteem and schizophrenia (how these topics and related with each other).
4.) In the Methods-chapter, it might be interesting to find more details about the program (e.g. the setup of the groups, weather conditions, ...)
5.) In the discussion-chapter, the results and their interpretations should then be linked in greater detail to the suggested additions of the Introducation-chapter.
6.) The last sentence of the Discussions-chapter is as follows: "Additionally, the possibility that interventions other than the agro-healing horticultural therapy program within the ward influenced the results cannot be completely ruled out.". Here, I would like to read some examples.
7.) Regarding the Literatur - some of it is rather old and there are not many recent references (from 2025). Could this fact be adapted?
Author Response
Reviewer 1
Dear Reviewer,
Thank you very much for your constructive review and excellent suggestions for our manuscript.
We have carefully considered all the comments and revised our manuscript accordingly. We have addressed each comment in detail in the point-by-point response below, with our responses indicated in bold. Changes to the manuscript text have been highlighted in red for easy identification.
We believe the manuscript has been significantly improved based on your valuable feedback. We sincerely hope that the revised manuscript is now suitable for publication in your journal.
Sincerely,
Thank you very much for this interesting article and your reserach about this important topic. To optimize this article, I would recommend following steps:
1.) Include short definitions of horticultural therapy and agro-healing horticultural therapy in the Introduction-chapter.
: We have revised the manuscript as requested. We have added and organized the short definitions of horticultural therapy and agro-healing horticultural therapy in the Introduction chapter (lines 56-74).
2.) Further develop the links and relationships between stress, self-esteem and schizophrenia in the Introduction-chapter and offer a short definition of stress and self-esteem (its meaning and the possibilities to measure it).
: We have further developed the links and relationships between stress, self-esteem, and schizophrenia in the Introduction, as requested in lines 82-117.
3.) Also in the Introduction-chapter, you write, that your aim was as follows: "The aim of the agro-healing program was to promote health restoration and rehabilitation through agricultural activities. ". Thus, it would be important to explain the connections between health restoration (and its promotion) and rehabilitations, stress, self-esteem and schizophrenia (how these topics and related with each other).
: We changed the sentence according to your suggestion in line 82-117
4.) In the Methods-chapter, it might be interesting to find more details about the program (e.g. the setup of the groups, weather conditions, ...)
: We have revised the Methods chapter to include more details about the program implementation, as requested in lines 134-244 of the revised manuscript.
5.) In the discussion-chapter, the results and their interpretations should then be linked in greater detail to the suggested additions of the Introduction-chapter.
: We have significantly revised the Discussion based on your feedback.
6.) The last sentence of the Discussions-chapter is as follows: "Additionally, the possibility that interventions other than the agro-healing horticultural therapy program within the ward influenced the results cannot be completely ruled out.". Here, I would like to read some examples.
: We have revised the final sentence of the Discussion to include specific examples of other potential interventions within the ward (e.g., music therapy, exercise therapy, art therapy, and social skill training) that might have influenced the results, as requested.
7.) Regarding the Literature - some of it is rather old and there are not many recent references (from 2025). Could this fact be adapted?
: We appreciate the observation regarding the age of some references. We have thoroughly updated the Literature section, integrating more recent publications (especially from 2025 and recent years) to ensure the discussion reflects the current state of research. The reference list and corresponding in-text citations have been adapted.
Reviewer 2 Report
Comments and Suggestions for Authors
Dear Authors,
Thank you for submitting your manuscript. We greatly appreciate the effort and dedication evident in exploring horticultural therapy as a supportive psychosocial intervention for inpatients with schizophrenia. Your study addresses an important and timely topic with clear potential for impact.
After a careful review, we have identified several areas that would benefit from substantial revision to enhance clarity, scientific accuracy, and alignment with publication standards. Please see the attached evaluation for detailed feedback. We encourage you to carefully consider these recommendations, as addressing them will strengthen the manuscript and its contribution to the field.
We are confident that, with careful revision, your work can be significantly improved and better highlight the promising findings of your study. .
Kind regards,
--------------------------
Title and Abstract: In my view, the title and abstract should make it clear that horticultural therapy is applied as a supportive psychosocial intervention targeting stress, self-esteem, and positive and negative symptoms in inpatients with schizophrenia, not as a curative treatment for schizophrenia itself.
Lines 38–45: This text presents a conceptual and scientific writing issue. It states that: “…horticultural therapy […] has been adopted […] for the treatment of schizophrenia” and “…interventions based on horticultural therapy are more effective than standard treatments for schizophrenia.” This implies that horticultural therapy treats schizophrenia itself, almost as if it were a primary or alternative therapeutic approach. However, schizophrenia is a chronic disorder with multiple dimensions (biological, psychological, social), and horticultural therapy is used as a complementary or rehabilitation intervention, not as an etiological or primary treatment.
Lines 65–66 and 74–76: These sentences are repetitive. I suggest removing lines 65–66 and expanding the paragraph to justify the relationship with stress, self-esteem, and positive and negative symptoms, thereby supporting the subsequent use of the PSS, SES, and PANSS.
Methods section:
Points 2.1 (Purpose) and 2.3 (Research Hypotheses) should be moved to the end of the introduction, either without a heading or under the heading “The present study” (in which case point 2.2, Design, should also be included).
In the participants section, it states: “The final sample consisted of 26 participants in the experimental group and 23 in the control group, meeting the required sample size.” In my opinion, this sample size is appropriate for a pilot or exploratory study. The data can help identify trends and potential effects of horticultural therapy, which could support hypotheses for larger studies. With this sample size, clinical efficacy cannot be claimed. Statistically, the probability of detecting a true effect depends on the expected effect size, data variability, and sample size. With small samples, typically only moderate-to-large effects are detectable; small effects may go undetected, increasing the risk of Type II errors. Confidence intervals will likely be wide, reducing the precision of effect estimates. Therefore, generalising results to the wider schizophrenia population is difficult. I recommend reporting results as promising but preliminary, and discussing the sample size limitation in a Limitations section.
Lines 96 and following: There is a mix of procedure description and participants information. Please create a section 2.1 Participants specifying sample size, mean age with standard deviation or range, gender percentages, and any other available sociodemographic information, or justify the absence of these data. Also, if the study employed a non-equivalent control group, this should be explained and justified.
Then, create a 2.2 Procedure section, orderly moving information currently scattered across other sections. Typically, this section should start with: Type of study conducted. Study period and location. Data collection, participant selection, inclusion/exclusion criteria. Program followed by experimental and control groups. How questionnaires were administered and data collected. Ethical considerations
Lines 124 and following – Ethical Considerations: It states that participants gave informed consent, but it is not mentioned whether decision-making capacity was formally assessed. In studies with adults with mental disorders, many ethical guidelines (Helsinki Declaration, CIOMS) recommend assessing each patient’s capacity before requesting informed consent. If there is doubt, consent should first be obtained from a legal representative.
Lines 134 and following – Measures: Point 2.6 should be renumbered 2.3 Measures. Measures should include: number of items, factorial structure (one factor, two factors, etc.), example items per factor, Likert scale with range, and Cronbach’s α obtained from the current study (bibliographic values are not sufficient). If α is inadequate, this should be justified and referenced in Limitations.
Line 174 and following. The point 2.7 Gardening Therapy should become part of 2.2 Procedure.
Lines 180–183 should be removed. Procedures should be described objectively, without conclusions or assumptions.
Table 1 – Agro-healing horticultural therapy programme: What is the source of this table? The control group is described as engaging in ward-based programmes (music therapy, exercise therapy, art therapy). Does the experimental group also follow this programme? Is horticultural therapy complementary or substitutive? If substitutive, two different therapies are being compared, which needs clarification.
Suggestion: In tables, font size can be reduced (11 pt instead of 12 pt), and brief descriptions may be included in the table to avoid lengthy post-table explanations.
Lines 271 and following – Section 2.8 Data Collection: This information should be moved to the beginning of Section 2.2 Procedure. It should be noted that this information has already appeared previously. Please avoid repetition.
Line 294 and following – Results, Section 3.1 Homogeneity Test of General Characteristics: Do not repeat in the text the data already presented in the table, unless it is a particularly significant figure necessary for the accompanying explanation. Table 2 includes age, sex, and BMI. Neither the Introduction nor the Participants section justifies the inclusion of BMI, an unusual parameter, unless it is relevant, in which case the reason should be clearly stated.
According to Table 3, the groups are only comparable in terms of negative symptoms.
Line 315, Section 3.2 Hypothesis Testing: The hypotheses were already stated in Section 2.3. They serve to address the study objectives and determine the statistical analyses to be performed. It is unclear why new hypotheses are being introduced in the Results section. Furthermore, on what basis is it stated that the perceived stress scores in the experimental group decreased from 17.67 to 15.74, while the control group decreased from 19.08 to 17.54, and similarly for the other measures?
The sample size is very small, so it is normal that homogeneity results between groups may not be satisfactory. I believe the objectives and approach should be reformulated based on this fact.
It is unclear whether the Discussion addresses the hypotheses from Section 2.3 or those from Section 3.2. Note that the Discussion should focus on whether the study objectives have been achieved and whether the results are consistent with expectations and references in the Introduction. It is not appropriate to introduce new data or references in the Discussion section.
The Discussion should be carefully reviewed to remove references that could suggest horticultural therapy could replace antipsychotic treatments. The basis of schizophrenia is more complex. The therapy analysed here aims to improve quality of life, but there is no scientific evidence that it can replace medication.
A Limitations section is missing, where the small sample size and other limitations should be discussed.
Lines 430 and following – Conclusion: Review and omit unverified statements. A study with such a small sample cannot confirm the efficacy of a programme for health improvement.
Author Response
Reviewer 2
Dear Reviewer,
Thank you very much for your constructive review and excellent suggestions for our manuscript.
We have carefully considered all the comments and revised our manuscript accordingly. We have addressed each comment in detail in the point-by-point response below, with our responses indicated in bold. Changes to the manuscript text have been highlighted in red for easy identification.
We believe the manuscript has been significantly improved based on your valuable feedback. We sincerely hope that the revised manuscript is now suitable for publication in your journal.
Sincerely,
Title and Abstract: In my view, the title and abstract should make it clear that horticultural therapy is applied as a supportive psychosocial intervention targeting stress, self-esteem, and positive and negative symptoms in inpatients with schizophrenia, not as a curative treatment for schizophrenia
itself.
: We have revised the title and abstract as requested.
TITLE: Effects of an Agro-healing Horticultural Intervention on Stress, Self-Esteem, and PANSS Scores in Schizophrenic Inpatients: : A Quasi-Experimental Study
Lines 38–45: This text presents a conceptual and scientific writing issue. It states that: “…horticultural therapy […] has been adopted […] for the treatment of schizophrenia” and “…interventions based on horticultural therapy are more effective than standard treatments for schizophrenia.” This implies that horticultural therapy treats schizophrenia itself, almost as if it were a primary or alternative therapeutic approach. However, schizophrenia is a chronic disorder with multiple dimensions (biological, psychological, social), and horticultural therapy is used as a complementary or rehabilitation intervention, not as an etiological or primary treatment.
: As requested, we have revised all the introduction sections.
Lines 65–66 and 74–76: These sentences are repetitive. I suggest removing lines 65–66 and expanding the paragraph to justify the relationship with stress, self-esteem, and positive and negative symptoms, thereby supporting the subsequent use of the PSS, SES, and PANSS.
: As requested, We have revised all the introduction sections. (Line 82-117)
Methods section:
Points 2.1 (Purpose) and 2.3 (Research Hypotheses) should be moved to the end of the introduction, either without a heading or under the heading “The present study” (in which case point 2.2, Design, should also be included).
: We have updated and revised the content in accordance with your suggestions.
In the participants section, it states: “The final sample consisted of 26 participants in the experimental group and 23 in the control group, meeting the required sample size.” In my opinion, this sample size is appropriate for a pilot or exploratory study. The data can help identify trends and potential effects of horticultural therapy, which could support hypotheses for larger studies. With this sample size, clinical efficacy cannot be claimed. Statistically, the probability of detecting a true effect depends on the expected effect size, data variability, and sample size. With small samples, typically only moderate-to-large effects are detectable; small effects may go undetected, increasing the risk of Type II errors. Confidence intervals will likely be wide, reducing the precision of effect estimates. Therefore, generalising results to the wider schizophrenia population is difficult. I recommend reporting results as promising but preliminary, and discussing the sample size limitation in a Limitations section.
: We sincerely appreciate the insightful comments regarding the sample size and the interpretation of results. We have carefully revised the manuscript based on your concerns to ensure the scope of the study is interpreted appropriately. In the limitations section, we added sample size limitations and conclusions.
Lines 96 and following: There is a mix of procedure description and participants information. Please create a section 2.1 Participants specifying sample size, mean age with standard deviation or range, gender percentages, and any other available sociodemographic information, or justify the absence of these data. Also, if the study employed a non-equivalent control group, this should be explained and justified.
Then, create a 2.2 Procedure section, orderly moving information currently scattered across other sections. Typically, this section should start with: Type of study conducted. Study period and location. Data collection, participant selection, inclusion/exclusion criteria. Program followed by experimental and control groups. How questionnaires were administered and data collected. Ethical considerations
: We have fully reorganized the Methods chapter structure as suggested.
Lines 124 and following – Ethical Considerations: It states that participants gave informed consent, but it is not mentioned whether decision-making capacity was formally assessed. In studies with adults with mental disorders, many ethical guidelines (Helsinki Declaration, CIOMS) recommend assessing each patient’s capacity before requesting informed consent. If there is doubt, consent should first be obtained from a legal representative.:
: We confirm that all revisions have been implemented in accordance with the your suggestions on 2.2.3
Lines 134 and following – Measures: Point 2.6 should be renumbered 2.3 Measures. Measures should include: number of items, factorial structure (one factor, two factors, etc.), example items per factor, Likert scale with range, and Cronbach’s α obtained from the current study (bibliographic values are not sufficient). If α is inadequate, this should be justified and referenced in Limitations.
: We have updated and revised the content in accordance with your suggestions.
Line 174 and following. The point 2.7 Gardening Therapy should become part of 2.2 Procedure.
: All changes have been made in accordance with your requests.
Lines 180–183 should be removed. Procedures should be described objectively, without conclusions or assumptions. : We have removed it.
Table 1 – Agro-healing horticultural therapy programme: What is the source of this table? The control group is described as engaging in ward-based programmes (music therapy, exercise therapy, art therapy). Does the experimental group also follow this programme? Is horticultural therapy complementary or substitutive? If substitutive, two different therapies are being compared, which needs clarification.
Suggestion: In tables, font size can be reduced (11 pt instead of 12 pt), and brief descriptions may be included in the table to avoid lengthy post-table explanations.
: All requested modifications have been applied as specified in Section 2.2.4. Table 1 has also been modified.
Lines 271 and following – Section 2.8 Data Collection: This information should be moved to the beginning of Section 2.2 Procedure. It should be noted that this information has already appeared previously. Please avoid repetition.
: We made all the necessary changes, as requested.
Line 294 and following – Results, Section 3.1 Homogeneity Test of General Characteristics: Do not repeat in the text the data already presented in the table, unless it is a particularly significant figure necessary for the accompanying explanation. Table 2 includes age, sex, and BMI. Neither the Introduction nor the Participants section justifies the inclusion of BMI, an unusual parameter, unless it is relevant, in which case the reason should be clearly stated.
: All corrections have been applied. BMI has been removed.
According to Table 3, the groups are only comparable in terms of negative symptoms.
: We fully agree with the reviewer's point (Table 3). To statistically address these limitations, we reanalyzed the data using analysis of covariance (ANCOVA). For all outcome variables showing significant differences at baseline, we included pre-test scores as covariates in the model.
Line 315, Section 3.2 Hypothesis Testing: The hypotheses were already stated in Section 2.3. They serve to address the study objectives and determine the statistical analyses to be performed. It is unclear why new hypotheses are being introduced in the Results section. Furthermore, on what basis is it stated that the perceived stress scores in the experimental group decreased from 17.67 to 15.74, while the control group decreased from 19.08 to 17.54, and similarly for the other measures?
: We agree with the reviewer that the heading was structurally inappropriate. We have deleted the heading "3.2 Hypothesis Testing" and replaced it with a result-oriented title: "3.2 Effects of the Agro-healing Intervention"
We have also revised the phrasing of hypotheses.
The sample size is very small, so it is normal that homogeneity results between groups may not be satisfactory. I believe the objectives and approach should be reformulated based on this fact.
: Based on the reviewer's comments, the conclusion has been revised.
It is unclear whether the Discussion addresses the hypotheses from Section 2.3 or those from Section 3.2. Note that the Discussion should focus on whether the study objectives have been achieved and whether the results are consistent with expectations and references in the Introduction. It is not appropriate to introduce new data or references in the Discussion section.
The Discussion should be carefully reviewed to remove references that could suggest horticultural therapy could replace antipsychotic treatments. The basis of schizophrenia is more complex. The therapy analysed here aims to improve quality of life, but there is no scientific evidence that it can replace medication.
: We have revised most of the discussion section based on your feedback.
A Limitations section is missing, where the small sample size and other limitations should be discussed.
: We changed the sentence according to your suggestion.
Lines 430 and following – Conclusion: Review and omit unverified statements. A study with such a small sample cannot confirm the efficacy of a programme for health improvement.
: We changed the sentence according to your suggestion.
“Conclusions
This study provides preliminary evidence suggesting the potential benefits of the agro-healing horticultural therapy program as an adjunctive intervention for patients with schizophrenia who are currently undergoing pharmacological treatment. Rehabilitation for these patients must be approached from a multidisciplinary perspective, rather than being solely a matter for psychiatry. The agro-healing horticultural therapy program integrates such multidisciplinary elements and can serve as a rehabilitation intervention under deinstitutionalization policies.
However, given the exploratory nature of this research and the constraints inherent to the small sample size and single facility setting, the findings should be interpreted with caution. Thus, it is recommended that subsequent studies employ a more extensive and diverse sample, encompassing a wider range of regions, institutions, and settings, to validate these initial findings.”
Reviewer 3 Report
Comments and Suggestions for Authors
Although the study focuses on patients with schizophrenia and uses a representative sample, the manuscript does not explain the conceptual relevance between the disorder and the selected outcome measures. Relevant theoretical background and literature review in the introduction section is also not provided. Both the Introduction and Section 2.2 of the Methods should consider revision.
Lines 46–63, the distinctions and connections between the “agro-healing horticultural therapy program” and the broader categories of horticultural therapy or therapeutic horticulture are not explained and should be clarified.
Line 108: When reporting the G*Power 3.1.9 sample size calculation, the manuscript should also specify the statistical test or analysis model used for the estimation.
Line 164, Section 2.7: The conceptual relationship and distinctions between agro-healing horticultural therapy and gardening therapy need to be clearly described.
Lines 186–197: Please provide detailed information on the routine ward program.
In Line 271, Section 2.8, The manuscript only states that “pre-test was conducted on the project start date, and post-test at the end of the project,” without precise specification. Please indicate whether assessments occurred immediately before the intervention on the same day or at other times, where they were conducted, and provide equivalent details for the control group.
From a statistical perspective, although baseline differences between groups are non-significant, it should include baseline values as covariates in the analysis model.
Author Response
Reviewer 3
Dear Reviewer,
Thank you very much for your constructive review and excellent suggestions for our manuscript.
We have carefully considered all the comments and revised our manuscript accordingly. We have addressed each comment in detail in the point-by-point response below, with our responses indicated in bold. Changes to the manuscript text have been highlighted in red for easy identification.
We believe the manuscript has been significantly improved based on your valuable feedback. We sincerely hope that the revised manuscript is now suitable for publication in your journal.
Sincerely,
Although the study focuses on patients with schizophrenia and uses a representative sample, the manuscript does not explain the conceptual relevance between the disorder and the selected outcome measures. Relevant theoretical background and literature review in the introduction section is also not provided. Both the Introduction and Section 2.2 of the Methods should consider revision.
: We agree with the reviewer that the theoretical and conceptual links between schizophrenia and the selected outcome measures (Stress, Self-Esteem, and PANSS) needed to be more thoroughly established.
Lines 46–63, the distinctions and connections between the “agro-healing horticultural therapy program” and the broader categories of horticultural therapy or therapeutic horticulture are not explained and should be clarified.
: We changed the sentence according to your suggestion.
Line 108: When reporting the G*Power 3.1.9 sample size calculation, the manuscript should also specify the statistical test or analysis model used for the estimation.
: We have made revisions based on your feedback. The statistical test used for the G*Power calculation has been specified as the F-test for Analysis of Covariance.
Line 164, Section 2.7: The conceptual relationship and distinctions between agro-healing horticultural therapy and gardening therapy need to be clearly described.
: We changed the sentence according to your suggestion.
Lines 186–197: Please provide detailed information on the routine ward program.
: We changed the sentence according to your suggestion.
In Line 271, Section 2.8, The manuscript only states that “pre-test was conducted on the project start date, and post-test at the end of the project,” without precise specification. Please indicate whether assessments occurred immediately before the intervention on the same day or at other times, where they were conducted, and provide equivalent details for the control group.
: As you requested, I have added the content in 2.2.2 Data collection
From a statistical perspective, although baseline differences between groups are non-significant, it should include baseline values as covariates in the analysis model.
: I have revised all the content by conducting an ANCOVA as per your suggestion.
Reviewer 4 Report
Comments and Suggestions for Authors
I have read with great interest the manuscript entitled “Effects of an Agro-Healing Horticultural Therapy Program on Patients with Schizophrenia: A Quasi-Experimental Study” and I congratulate the authors for their initiative and efforts. The manuscript approaches an important aspect of the treatment of patients with schizophrenia, namely rehabilitation. However, the manuscript needs major revision in my opinion. My comments and suggestions for improvement of the manuscript are presented below:
Introduction
Lines 56-59: this paragraph should not be included in the introduction; it describes the agro-healing therapy program and its rationale therefore it belongs to the materials and methods section.
Line 38: this line should be in the final part of the introduction section, in the paragraph describing the aims and objectives of the study.
Lines 65-76: This section needs to be rewritten and should contain only the aim and objectives of the study.
The introduction reiterates the study aim multiple times (e.g., “this study aims to implement…” appears twice). It would read more fluently if the aim were stated only once at the end of the section, following a logical progression if the introduction section: context – rationale – evidence base – gap – study aim. Streamline redundant sentences and emphasize the gap in the literature — e.g., the limited number of controlled studies testing agro-healing therapy for schizophrenia, particularly with validated psychological outcomes (stress, PANSS, self-esteem).
The introduction lists multiple reported benefits of horticultural therapy (improved confidence, reduced anxiety, enhanced social functioning), but it would be useful to explicitly connect these expected outcomes to the measured variables in this study (perceived stress, PANSS, self-esteem). The authors should include information regarding perceived stress and self-esteem in schizophrenia patients outlying the importance of improving them. Add a bridging paragraph summarizing why these particular outcomes were chosen and how they reflect key domains of recovery in schizophrenia.
Moreover, even if the objective of the study is not to measure quality of life, reducing perceived stress and increasing self-esteem could improve quality of life which is an important outcome for psychiatric interventions. While the introduction repeatedly mentions “quality of life,” it does not theoretically define or operationalize this construct. Given that QoL is central to schizophrenia research, the authors should briefly outline its psychosocial determinants (e.g., self-esteem, social support) and its clinical correlates (e.g., symptom severity, functioning). Expand the introductory discussion on QoL by integrating recent findings on its psychosocial predictors. For exemple, this study - https://doi.org/10.3390/jcm13226959 - would be particularly relevant. It highlights how self-esteem and social support (both targeted by agro-healing interventions) are crucial to improving QoL in schizophrenia.
Materials and methods:
Lines 189-270: I suggest to the authors to include this information as Supplementary Material; some key information from the description of each session should be included in Table 1. The authors could add a different column presenting separately the objective of each statement by dividing the column “hey factors“ in two: activities and session objectives and outcomes. Therefore, I suggest keeping only Table 1 but after improvements.
Lines 79-81: the purpose of the study should be included in the last paragraph from the introduction (the one describing aim and objectives).
Lines 272-281: I think this paragraph needs more details. How were the patients selected for inclusion in this study? Were all the patients from the 2 wards were selected? Was the project presented to all the patients and then all volunteers satisfying the inclusion and exclusion criteria were included? Who did the selection and how? A flow-diagram with the selection process should be included.
The authors should specify how often were the sessions scheduled. Weekly? Twice a week? How long did the program last? In data collection section the authors mentioned from August to October only.
Results:
Lines 295-304: this paragraph should be shortened keeping only the essential information (information described here is already included in the table 2).
Lines 283-292, 328-329: The authors report that pre- and post-test scores in both the experimental and control groups were compared using the Wilcoxon signed-rank test. However, this test only assesses within-group changes and does not evaluate between-group differences and I believe that the statistical analysis should be improved. Since the two groups differed significantly at baseline for the main outcome variables (perceived stress, PANSS, and self-esteem), using within-group tests alone is insufficient to support the conclusions about the intervention’s efficacy in the area of PANSS, self-esteem and perceived stress. I suggest comparing the change scores (Delta post–pre) between groups using a Mann–Whitney U test or conducting an ANCOVA controlling for baseline differences. Unfortunately, the conclusion that agro-healing therapy was effective compared to the control condition is not statistically substantiated. The authors must revised this part and apl=ply the appropriate statistical analysis.
Discussion:
The discussion should be changed if the revised statistical analysis shows different results.
It will be useful to discuss whether the duration or intensity of the intervention was sufficient to produce psychological changes or whether self-esteem in schizophrenia may be more resistant to short-term intervention due to its association with chronic internalized stigma and social withdrawal.
Lines 379-381: the authors should expand the discussion and explore how the agro-healing horiculture therapy could lower the PANSS scores. I believe that the effect is indirect and the authors should provide an explanation.
Ensure uniform terminology (“agro-healing horticultural therapy” vs. “horticultural therapy”).
The limitation paragraph should incorporate the small number of patients included, the heterogenity between the 2 group at pre-exposure time point, that fact that the participants were only inpatients. The conclusion paragraph should be revised accordingly.
Conclusions:
Lines 431-432: it was not specified in the Materials and Methods section that the investigators chose patients with a high dependence on medication. This conclusion might be misleading. Also, the authors should explicity state the domains in which the program was efficient and that the participants were inpatients otherwise this statement is too general and therefore also misleading.
Author Response
Reviewer 4
Dear Reviewer,
Thank you very much for your constructive review and excellent suggestions for our manuscript.
We have carefully considered all the comments and revised our manuscript accordingly. We have addressed each comment in detail in the point-by-point response below, with our responses indicated in bold. Changes to the manuscript text have been highlighted in red for easy identification.
We believe the manuscript has been significantly improved based on your valuable feedback. We sincerely hope that the revised manuscript is now suitable for publication in your journal.
Sincerely,
Introduction
Lines 56-59: this paragraph should not be included in the introduction; it describes the agro-healing therapy program and its rationale therefore it belongs to the materials and methods section.
: We have implemented the requested structural change. We confirm that the paragraph detailing the description and rationale of the agro-healing horticultural therapy program has been deleted from the introduction section
Line 38: this line should be in the final part of the introduction section, in the paragraph describing the aims and objectives of the study.
: We changed the sentence according to your suggestion.
Lines 65-76: This section needs to be rewritten and should contain only the aim and objectives of the study. : We changed the sentence according to your suggestion.
The introduction reiterates the study aim multiple times (e.g., “this study aims to implement…” appears twice). It would read more fluently if the aim were stated only once at the end of the section, following a logical progression if the introduction section: context – rationale – evidence base – gap – study aim. Streamline redundant sentences and emphasize the gap in the literature — e.g., the limited number of controlled studies testing agro-healing therapy for schizophrenia, particularly with validated psychological outcomes (stress, PANSS, self-esteem).
: We have revised all the contents of the introduction you pointed out and marked them in red.
The introduction lists multiple reported benefits of horticultural therapy (improved confidence, reduced anxiety, enhanced social functioning), but it would be useful to explicitly connect these expected outcomes to the measured variables in this study (perceived stress, PANSS, self-esteem). The authors should include information regarding perceived stress and self-esteem in schizophrenia patients outlying the importance of improving them. Add a bridging paragraph summarizing why these particular outcomes were chosen and how they reflect key domains of recovery in schizophrenia.
: All revisions have been completed in accordance with your requests.
Moreover, even if the objective of the study is not to measure quality of life, reducing perceived stress and increasing self-esteem could improve quality of life which is an important outcome for psychiatric interventions. While the introduction repeatedly mentions “quality of life,” it does not theoretically define or operationalize this construct. Given that QoL is central to schizophrenia research, the authors should briefly outline its psychosocial determinants (e.g., self-esteem, social support) and its clinical correlates (e.g., symptom severity, functioning). Expand the introductory discussion on QoL by integrating recent findings on its psychosocial predictors. For exemple, this study - https://doi.org/10.3390/jcm13226959 - would be particularly relevant. It highlights how self-esteem and social support (both targeted by agro-healing interventions) are crucial to improving QoL in schizophrenia.
: As you requested, we have revised it.
Materials and methods:
Lines 189-270: I suggest to the authors to include this information as Supplementary Material; some key information from the description of each session should be included in Table 1. The authors could add a different column presenting separately the objective of each statement by dividing the column “hey factors“ in two: activities and session objectives and outcomes. Therefore, I suggest keeping only Table 1 but after improvements.
: We changed the sentence according to your suggestion.
Table 1 has been separated into Appendix 1
.
Lines 79-81: the purpose of the study should be included in the last paragraph from the introduction (the one describing aim and objectives).
: : We changed the sentence according to your suggestion.
Lines 272-281: I think this paragraph needs more details. How were the patients selected for inclusion in this study? Were all the patients from the 2 wards were selected? Was the project presented to all the patients and then all volunteers satisfying the inclusion and exclusion criteria were included? Who did the selection and how? A flow-diagram with the selection process should be included.
: We changed the sentence according to your suggestion on 2.1. Participants.
The authors should specify how often were the sessions scheduled. Weekly? Twice a week? How long did the program last? In data collection section the authors mentioned from August to October only.
: We changed the sentence according to your suggestion on 2.2.4. Agro-healing horticulture therapy
Results:
Lines 295-304: this paragraph should be shortened keeping only the essential information (information described here is already included in the table 2).
: We changed the sentence according to your suggestion
Lines 283-292, 328-329: The authors report that pre- and post-test scores in both the experimental and control groups were compared using the Wilcoxon signed-rank test. However, this test only assesses within-group changes and does not evaluate between-group differences and I believe that the statistical analysis should be improved. Since the two groups differed significantly at baseline for the main outcome variables (perceived stress, PANSS, and self-esteem), using within-group tests alone is insufficient to support the conclusions about the intervention’s efficacy in the area of PANSS, self-esteem and perceived stress. I suggest comparing the change scores (Delta post–pre) between groups using a Mann–Whitney U test or conducting an ANCOVA controlling for baseline differences. Unfortunately, the conclusion that agro-healing therapy was effective compared to the control condition is not statistically substantiated. The authors must revised this part and apply the appropriate statistical analysis.
: We fully agree with the statistical recommendation regarding baseline differences.
: changed ANCOVA
Discussion:
The discussion should be changed if the revised statistical analysis shows different results.
It will be useful to discuss whether the duration or intensity of the intervention was sufficient to produce psychological changes or whether self-esteem in schizophrenia may be more resistant to short-term intervention due to its association with chronic internalized stigma and social withdrawal.
: We confirm that we conducted the re-analysis using the Analysis of Covariance (ANCOVA) as requested. The revised statistical analysis yielded results consistent with our initial findings regarding the significance and direction of the intervention effects. Specifically, ANCOVA confirmed that the adjusted post-test means for the experimental group were significantly different from the control group in the predicted directions for our primary outcome measures.
Lines 379-381: the authors should expand the discussion and explore how the agro-healing horiculture therapy could lower the PANSS scores. I believe that the effect is indirect and the authors should provide an explanation.
: We have carefully reviewed all comments and have revised the discussion section accordingly.
Ensure uniform terminology (“agro-healing horticultural therapy” vs. “horticultural therapy”).
: We have made the revisions according to your feedback.
The limitation paragraph should incorporate the small number of patients included, the heterogenity between the 2 group at pre-exposure time point, that fact that the participants were only inpatients. The conclusion paragraph should be revised accordingly.
: : We have revised the conclusion section based on your comments.
“This study's results must be interpreted with caution due to several methodological constraints. First, the small number of patients included, coupled with the fact that participants were only inpatients from a single facility, limits the generalizability of the findings to diverse populations or settings (e.g., outpatient care). Second, there was a degree of baseline heterogeneity between the experimental and control groups at the pre-exposure time point. Finally, the possibility that other simultaneous interventions within the ward (e.g., music therapy, exercise therapy, art therapy, and social skill training) influenced the results cannot be completely ruled out.
However, the strength of this study lies in the fact that a horticultural therapist directly administered horticultural therapy to patients with schizophrenia, while a psychiatrist and mental health nurse provided support for symptom monitoring and assessment throughout the entire process. Considering the current reality that mental health professionals (e.g., psychiatrists and nurses) in psychiatric wards seek to implement agro-healing horticultural therapy for community reintegration, but face challenges due to the qualifications and prerequisites required to deliver the program [16], this study may be considered meaningful.
Despite these limitations, the present study offers valuable evidence supporting the efficacy and unique potential of agro-healing horticultural therapy as an adjunctive intervention for schizophrenia. Future research should address these methodological constraints to further validate the findings.”
Conclusions:
Lines 431-432: it was not specified in the Materials and Methods section that the investigators chose patients with a high dependence on medication. This conclusion might be misleading. Also, the authors should explicity state the domains in which the program was efficient and that the participants were inpatients otherwise this statement is too general and therefore also misleading.
: We have revised the conclusion section based on reviewer comments.
“This study provides preliminary evidence suggesting the potential benefits of the agro-healing horticultural therapy program as an adjunctive intervention for patients with schizophrenia who are currently undergoing pharmacological treatment. Rehabilitation for these patients must be approached from a multidisciplinary perspective, rather than being solely a matter for psychiatry. The agro-healing horticultural therapy program integrates such multidisciplinary elements and can serve as a rehabilitation intervention under deinstitutionalization policies.
However, given the exploratory nature of this research and the constraints inherent to the small sample size and single facility setting, the findings should be interpreted with caution. Thus, it is recommended that subsequent studies employ a more extensive and diverse sample, encompassing a wider range of regions, institutions, and settings, to validate these initial findings.”
Round 2
Reviewer 2 Report
Comments and Suggestions for Authors
First of all, I would like to congratulate the authors on the improvements incorporated into the manuscript; I believe it is now much stronger and could be considered for publication. However, I recommend the following minor modifications:
-
Lines 201 and following: The therapy is described in sufficient detail, but information is missing about who conducts it. What training do the individuals working with the patients have? Are they experts in the application of this therapy? What kind of training have they received?
-
Line 478: It is stated that “These effects are considered indirect.” It is unclear whether the authors consider the effect to be indirect or if this statement comes from the literature. This should be clarified.
-
Lines 479 and following: The attempt to propose a possible explanation is appreciated, but it is recommended to reformulate the text to provide a stronger consolidation of the hypothesis.
-
Line 483: It is suggested to use a paragraph break instead of a full stop.
-
Line 495: The fact that the results are statistically significant does not allow one to claim that the improvement is clinically significant. Please reformulate.
-
Line 506: It is recommended to change the expression “provided opportunities to improve” to “may provide opportunities to improve”. Similar expressions are found in subsequent sentences; it would be advisable to review the text and adjust any conclusions accordingly, considering that this is a quasi-experimental study with a very small sample.
Author Response
Dear Reviewer,
We are sincerely grateful for your thorough review and constructive comments on the second round of review for our manuscript. Your constructive feedback has been invaluable in significantly improving the clarity and overall quality of the manuscript.
We have carefully considered all your remaining comments and revised the manuscript accordingly. As shown below, we have addressed each comment in the point-by-point response, with our replies indicated in bold. Changes made to the manuscript text have been highlighted in red for easy identification.
Once again, we thank you for your dedication and support of our research.
Sincerely,
First of all, I would like to congratulate the authors on the improvements incorporated into the manuscript; I believe it is now much stronger and could be considered for publication. However, I recommend the following minor modifications:
- Lines 201 and following: The therapy is described in sufficient detail, but information is missing about who conducts it. What training do the individuals working with the patients have? Are they experts in the application of this therapy? What kind of training have they received?
Response: The revised text now includes the following details:
“The program delivery was subsequently entrusted to four agricultural researchers who served as the facilitators. These individuals possess substantial expertise in horticultural therapy, having each accumulated over ten years of experience in the field. In addition, they have each accumulated a minimum of three years' experience in planning and implementing these agro-healing programs specifically for patients in psychiatric wards.”
- Line 478: It is stated that “These effects are considered indirect.” It is unclear whether the authors consider the effect to be indirect or if this statement comes from the literature. This should be clarified.
Response: We have made the requested changes:
“As a non-pharmacological intervention, these effects are interpreted as indirect, contrasting with the direct biological action of medication.”
- Lines 479 and following: The attempt to propose a possible explanation is appreciated, but it is recommended to reformulate the text to provide a stronger consolidation of the hypothesis.
Response: We have made the requested changes:
“Active engagement in tasks such as planting or weeding is demonstrated to serve as a valuable form of behavioral distraction from positive symptoms.”
- Line 483: It is suggested to use a paragraph break instead of a full stop.
Response: The text has been reformulated by separating the two sentences into distinct
paragraphs, as suggested.
- Line 495: The fact that the results are statistically significant does not allow one to claim that the improvement is clinically significant. Please reformulate.
Response: We have made the requested changes:
“Specifically, these results strengthen the hypothesis that the substantial physical activity and interactive activities inherent in the intervention, conducted in an outdoor environment, were effective in promoting positive symptom relief.”
- Line 506: It is recommended to change the expression “provided opportunities to improve” to “may provide opportunities to improve”. Similar expressions are found in subsequent sentences; it would be advisable to review the text and adjust any conclusions accordingly, considering that this is a quasi-experimental study with a very small sample.
Response: The necessary revisions have been made based on your feedback.
Reviewer 3 Report
Comments and Suggestions for Authors
The authors have made substantial revisions, but several detailed issues still require further improvement:
- Lines 58–59: I do not agree that horticultural therapy can only be conducted indoors. It can also take place outdoors, such as in gardens, courtyards, or even nearby parks. Please provide a more accurate explanation of the relevant terminology in this field and avoid presenting a biased definition.
- Lines 143–145: The manuscript mentions “predefined inclusion and exclusion criteria,” yet these criteria only appear later in the paper. Please adjust the order so the reader can understand the study design more clearly.
- Lines 248–250: “The control group did not participate in the agro-healing program. Instead, they engaged in ward-based programs, including music therapy, exercise therapy, art therapy, and social skill training.”
While it is acceptable that the control group is not a blank group, the activities listed here seem highly heterogeneous and involve multiple complementary or alternative therapy types. I recommend that the authors describe the control condition in more detail so readers can understand how it differs from the experimental group, including whether the time schedule and session structure were equivalent. Without clearer information, the validity of the study’s conclusions may be difficult to assess. - Lines 348–354: The manuscript discusses reliability testing of the scales and cites other studies using the same measures, but it does not explain why the reliability coefficients in the present study are relatively low. Please provide an explanation for this discrepancy.
- Line 370: Please maintain consistent decimal places.
- Table 4: It is unclear which column the significance values refer to—Pretest, Posttest, or Difference. Please clarify this and make the table information complete.
Author Response
Reviewer 3
Dear Reviewer,
We are sincerely grateful for your thorough review and constructive comments on the second round of review for our manuscript. Your constructive feedback has been invaluable in significantly improving the clarity and overall quality of the manuscript.
We have carefully considered all your remaining comments and revised the manuscript accordingly. As shown below, we have addressed each comment in the point-by-point response, with our replies indicated in bold. Changes made to the manuscript text have been highlighted in red for easy identification.
Once again, we thank you for your dedication and support of our research.
Sincerely,
The authors have made substantial revisions, but several detailed issues still require further improvement:
- Lines 58–59: I do not agree that horticultural therapy can only be conducted indoors. It can also take place outdoors, such as in gardens, courtyards, or even nearby parks. Please provide a more accurate explanation of the relevant terminology in this field and avoid presenting a biased definition.
Response: The changes have been made as requested.
“This therapy is characterized by its flexible application in diverse environments, ranging from dedicated indoor spaces to various outdoor settings such as therapeutic gardens or courtyards [3, 9]. This adaptability allows participants to engage with plants and experience the natural changes of the four seasons.”
- Lines 143–145:The manuscript mentions “predefined inclusion and exclusion criteria,” yet these criteria only appear later in the paper. Please adjust the order so the reader can understand the study design more clearly.
Response: We have implemented the revisions as requested.
- Lines 248–250:“The control group did not participate in the agro-healing program. Instead, they engaged in ward-based programs, including music therapy, exercise therapy, art therapy, and social skill training.”
While it is acceptable that the control group is not a blank group, the activities listed here seem highly heterogeneous and involve multiple complementary or alternative therapy types. I recommend that the authors describe the control condition in more detail so readers can understand how it differs from the experimental group, including whether the time schedule and session structure were equivalent. Without clearer information, the validity of the study’s conclusions may be difficult to assess.
Response: We have revised the Procedures section to provide a more detailed description of the structure and content of the control condition. Furthermore, to ensure transparency in the interpretation of the findings, we added a discussion of the resulting disparity in total intervention sessions to the Limitations section.
“The control group did not participate in the main agro-healing intervention. Instead, they participated in a standard, ward-based psychiatric rehabilitation program that included music, exercise, and art therapies, as well as social skills training. The program was structured to meet four days a week to ensure equivalence in contact time. Both groups participated in the program for 50 minutes daily, Monday through Thursday. The control group's weekly schedule was fixed: music therapy on Monday, exercise therapy on Tuesday, art therapy on Wednesday, and social skills training on Thursday.
The Friday sessions varied between the two groups. Only the experimental group received the complementary agricultural healing horticultural therapy session on Friday. Thus, over the 10-week period, the control group received a total of 40 sessions (four times a week), while the experimental group received 50 sessions (five times a week). Importantly, the standard ward program for the control group did not include exposure to agricultural activities, which were a core element of the experimental program.”
Limitation: “Finally, the difference in the total intervention dosage is a key limitation. Given that the experimental group received an additional treatment session on Fridays (50 total sessions vs. 40 total sessions), the influence of this increased contact time and attention on the results cannot be entirely discounted. Future studies should address these constraints, particularly by considering designs that strictly equalize the total contact time across both groups.”
- Lines 348–354:The manuscript discusses reliability testing of the scales and cites other studies using the same measures, but it does not explain why the reliability coefficients in the present study are relatively low. Please provide an explanation for this discrepancy.
Response: We acknowledge the reviewer's observation regarding the slight difference in the Perceived Stress Scale (PSS) reliability coefficient (α = 0.75) compared to values reported in other studies (e.g., Harris et al.'s Cronbach’s α = 0.89). While the Cronbach’s α value of 0.75 meets the generally accepted standard for research purposes, we attribute this minor discrepancy primarily to the characteristics of our specific sample, rather than an inherent flaw in the instrument:
(1) Sample Homogeneity: Our participants are inpatients in specialized psychiatric wards. This sample is highly restricted and likely more homogeneous in terms of their acute stress levels and psychological state than the general population samples often used in large-scale validation studies. When a sample exhibits restricted variability (high homogeneity), the internal consistency coefficient (α) tends to be lower because the variance of the total score is reduced relative to the sum of item variances.
(2) Sample Size: Our study utilized a relatively small sample (N=49) compared to potentially massive samples used in reference studies. A smaller sample size increases the sampling variation of the Cronbach’s α coefficient, which can account for the observed difference from published norms.
We confirm that despite this contextual variation, the PSS demonstrated an acceptable level of internal consistency (Cronbach’s α 0.70) for this specific clinical population. We believe that these revisions have substantially improved the clarity, completeness, and academic rigor of our manuscript. We are deeply grateful for your thoughtful and constructive comments, which have been invaluable to the quality of our paper.
- Line 370:Please maintain consistent decimal places.
Response: We have reviewed the manuscript and ensured consistent use of decimal places throughout the text, tables, and figures, as requested.
- Table 4:It is unclear which column the significance values refer to—Pretest, Posttest, or Difference. Please clarify this and make the table information complete.
Response: We have fully addressed this important point to ensure the table information is complete and unambiguous for the reader. We clarified the meaning of the significance values in Table 4 by adding the following detailed footnote:
“Note: The F and p values were derived from the Analysis of Covariance (ANCOVA) controlling Pretest scores. The p value specifically indicates the statistical significance of the difference between the Experimental and Control groups at the Posttest.”
Reviewer 4 Report
Comments and Suggestions for Authors
I am satisfied with the changes made, and I appreciate that you have carefully addressed all of my comments and recommendations. The manuscript has been substantially improved during first round of revisions.
I have only one recommendation since the authors have made changes to the title during first round of revisions:
I recommend replacing the term “schizophrenic inpatients” in the title with “inpatients with schizophrenia”. Terms such as "schizophrenic inpatients” are considered stigmatizing according to current guidelines.
Author Response
Reviewer 4
Dear Reviewer,
We are sincerely grateful for your thorough review and constructive comments on the second round of review for our manuscript. Your constructive feedback has been invaluable in significantly improving the clarity and overall quality of the manuscript.
We have carefully considered all your remaining comments and revised the manuscript accordingly. As shown below, we have addressed each comment in the point-by-point response, with our replies indicated in bold. Changes made to the manuscript text have been highlighted in red for easy identification.
Once again, we thank you for your dedication and support of our research.
Sincerely,
I am satisfied with the changes made, and I appreciate that you have carefully addressed all of my comments and recommendations. The manuscript has been substantially improved during first round of revisions.
I have only one recommendation since the authors have made changes to the title during first round of revisions:
I recommend replacing the term “schizophrenic inpatients” in the title with “inpatients with schizophrenia”. Terms such as "schizophrenic inpatients” are considered stigmatizing according to current guidelines.
Response: We sincerely thank the reviewer for this highly important and insightful recommendation. We fully agree that person-first language must be prioritized.
Action Taken: We have immediately replaced the term "schizophrenic inpatients" with "inpatients with schizophrenia" in the title and throughout the entire manuscript to align with current ethical and professional guidelines, thereby avoiding stigmatizing language.
The revised title is now: "Effects of an Agro-healing Horticultural Intervention on Stress, Self-Esteem, and PANSS Scores in Inpatients with Schizophrenia: A Quasi-Experimental Study."